# Flax *(Linum usitatissimum* L.): A Potential Candidate for Phytoremediation? Biological and Economical Points of View

**DOI:** 10.3390/plants9040496

**Published:** 2020-04-13

**Authors:** Muhammad Hamzah Saleem, Shafaqat Ali, Saddam Hussain, Muhammad Kamran, Muhammad Sohaib Chattha, Shoaib Ahmad, Muhammad Aqeel, Muhammad Rizwan, Nada H. Aljarba, Saad Alkahtani, Mohamed M. Abdel-Daim

**Affiliations:** 1MOA Key Laboratory of Crop Ecophysiology and Farming System in the Middle Reaches of the Yangtze River, College of Plant Science and Technology, Huazhong Agricultural University, Wuhan 430070, China; saleemhamza312@webmail.hzau.edu.cn (M.H.S.); sohaib.psp@webmail.hzau.edu.cn (M.S.C.); shoaib17@webmail.hzau.edu.cn (S.A.); 2Department of Environmental Sciences and Engineering, Government College University Allama Iqbal Road, Faisalabad 38000, Pakistan; mrazi1532@yahoo.com; 3Department of Biological Sciences and Technology, China Medical University, Taichung 40402, Taiwan; 4Department of agronomy, University of Agriculture, Faisalabad 38040, Punjab, Pakistan; shussain@uaf.edu.pk; 5Key Laboratory of Arable Land Conservation (Middle and Lower Reaches of Yangtze River), Ministry of Agriculture, Huazhong Agricultural University, Wuhan 430070, China; kamiagrarian763@gmail.com; 6State Key laboratory of Grassland Agro-Ecosystems, School of Life Science, Lanzhou University, Lanzhou 73000, China; aqeelbutt99@gmail.com; 7Department of Biology, College of Science, Princess Nourah bint Abdulrahman University, Riyadh 11474, Saudi Arabia; Nhaljarba@pnu.edu.sa; 8Department of Zoology, College of Science, King Saud University, Riyadh 11451, Saudi Arabia; salkahtani@ksu.edu.sa (S.A.); abdeldaim.m@vet.suez.edu.eg (M.M.A.-D.); 9Pharmacology Department, Faculty of Veterinary Medicine, Suez Canal University, Ismailia 41522, Egypt

**Keywords:** *Linum usitatissimum* (flax), heavy metals, phytoremediation, chelating agents, environmental pollution

## Abstract

Flax (*Linum usitatissimum* L.) is an important oil seed crop that is mostly cultivated in temperate climates. In addition to many commercial applications, flax is also used as a fibrous species or for livestock feed (animal fodder). For the last 40 years, flax has been used as a phytoremediation tool for the remediation of different heavy metals, particularly for phytoextraction when cultivated on metal contaminated soils. Among different fibrous crops (hemp, jute, ramie, and kenaf), flax represents the most economically important species and the majority of studies on metal contaminated soil for the phytoextraction of heavy metals have been conducted using flax. Therefore, a comprehensive review is needed for a better understanding of the phytoremediation potential of flax when grown in metal contaminated soil. This review describes the existing studies related to the phytoremediation potential of flax in different mediums such as soil and water. After phytoremediation, flax has the potential to be used for additional purposes such as linseed oil, fiber, and important livestock feed. This review also describes the phytoremediation potential of flax when grown in metal contaminated soil. Furthermore, techniques and methods to increase plant growth and biomass are also discussed in this work. However, future research is needed for a better understanding of the physiology, biochemistry, anatomy, and molecular biology of flax for increasing its pollutant removal efficiency.

## 1. Introduction

Most heavy metals are produced by pollution and their presence causes many ecological, evolutionary, and nutritional problems. Many risks are created due to heavy metal contamination, like soil pollution as well as security of food and its quality [1,2,3,4,5,6]. Heavy metals cause many harmful effects for living things, including plants. They decrease the growth and development of plants even at low concentrations of heavy metals with respect to other metals. Excess amounts of other metals or elements do not damage the tissues/cells of the plant, and their accumulation can even increase the growth of the plant [7,8,9,10,11]. The metals which are lethal or harmful for plants include: lead, cadmium, cobalt, iron, silver, platinum, nickel, chromium, copper, and zinc. Agricultural soil is befouled by many heavy metals, which is a major issue. Many anthropogenic activities effect soil and the effectiveness of the plant, like urbanization, smelting, sludge, military operations, mining, dumping, and excess amounts of pesticide and insecticide applications [3,5,10,12,13,14]. These activities inhibit the growth of the plants and cause many toxic symptoms in the plants due to heavy metal contamination, and these metals destroy human health by entering into the food chain [15,16,17]. However, the metal contaminants are useful in fertilization as well as in pesticides, fungicides, and nematicides for the better production and development of the plant. These applications are a major source for the growth of the plants. Metal deposition in soil is produced due to biomass and an increase in plant growth [3,18].

The chemical composition, plant type and their species, and the pH of heavy metals all impact the phytotoxicity of the heavy metals in the plants. A large amount of reacting oxygen species and oxidative stress are involved through direct and indirect toxicity of heavy metals and produced in the following ways: (a) reduction of electrons, (b) the irregular management of metabolic pathways, (c) the activities of anti-oxidative enzymes are reduced, and (d) depletion of lower molecular weight compounds [19,20,21,22]. Crop yield and productivity can be improved and the health risk decreased acutely by removing the harmful effects of these heavy metals [23,24,25].

For the removal of toxic heavy metals from the soil, two methods are usually used. The first is the physical–chemical technology method and the second is the conventional method. These methods are not widely used and may be costly because these methods are not eco-friendly. Some factors involved in these methods include: vapor extraction, stabilization, solidification, soil flushing, soil washing, thermal desorption, and incineration, etc. Some drawbacks are also involved in these methods. For example, they require constant monitoring, exact results through keen performance, and a large monetary cost. Through these methods, pollutants can be removed but harmful effects can be created for the environment due to spreading of pollutants, excavation, and handling, which are toxic [26,27]. Along with these drawbacks, one of the most effective, cheap, and environmental-friendly methods used for the removal of toxic substances which accumulate in the different parts of the plants, like in roots, leaves, branches, and stems, is known as phytoremediation [28]. This method is also used for the reduction of the metal content of soil, the assembly of very large amounts of toxic metal contents through repeating harvesting, and the cycling of planting of the plant species [29,30]. The phytoremediation mechanism is species-specific, which is mostly built upon some anatomical and morphological as well as physiological characteristics of the plants [4,26,31,32,33,34,35]. Phytoremediation is a technology which is widely applicable for metal contaminated areas, with some long-term aesthetic merits and it is famous due to its low cost and eco-friendly nature, so it is used on large scale areas with high contents of heavy metals due to some industry or mining activities.

Another method for the removal of toxic substances like heavy metals is “hyper accumulation,” which is widely used and uses the genetic potential of the roots of plants originating from contaminated areas [9,36]. Different plants have different methods for the removal and accumulation of different substances like nutrients and organic materials. For example, some plants have the ability to immobilize or absorb metal pollutants, and some plants can metabolize and accumulate organic species and nutrient contaminants. Many phytoremediation processes are possible through better relationships in between plants, microbes, soil, and contaminants. There are five different methods of utilizing the phytoremediation: (1) phytotransformation, (2) rhizosphere bioremediation, (3) phytostabilization, (4) phytoextraction, and (5) rhizofiltration. These five different processes of phytoremediation perform different functions and are different management options for better end products [13,26]. The mechanisms of different types of phytoremediation in plants with different management options of the end product are presented in Figure 1.

Hyper accumulator plants play a significant role in accumulating toxic heavy metals and absorb them in their roots and aboveground parts, which demonstrates that plants have the genetic potential to remove these toxic heavy metals from metal contaminated areas [15,37,38,39]. Phytoextraction is the most common method of contaminant removal, in which plant species have a huge biomass and accumulate a large amount of heavy metals in their above ground parts. This method depends on the abnormal capability of some plant species in accumulating heavy metals, and heavy metal tolerance is essential for plants used in phytoremediation methods. It is very important to select a plant species with a large biomass and great ability to accumulate heavy metals from the soil [31]. It is well known that heavy metals such as Cd, Ni, Zn, As, Se, and Cu are readily bioavailable for plants while Co, Mn, and Fe are moderately available, and Pb, Cr, and U are not easily available for plants. Pb, Cr, and U can be removed by binding to soils and root masses via rhizofiltration [29]. According to the different available literature on phytoremediation, we can say the purposes of phytoremediation are (i) risk containment, (ii) phytoextraction of metals with market value, such as Ni, Tl, and Au, and (iii) durable land management [29,40].

Flax (*Linum usitatissimum* L.), also known as common flax or linseed, is a fibrous crop and dicotyledonous plant belonging to the family Linaceae with potential economic value [12,41]. Flax is a bluish-flowered plant and has been used for fiber and food purposes in mainly the cooler regions of the world. It is an annual, herbaceous plant found significantly in many countries like China, Russia, and Canada. Flax is one of the oldest fibrous crops, and has been cultivated in China and India for 5000 years and was cultivated in Egypt and Samaria 10,000 years ago [42,43]. The first cultivations of flax in European countries were in Switzerland, Scotland, and Poland. Flax is also grown as an ornamental plant in gardens because of its attractive flowers. Flax has been used as fiber and also for the flaxseed. Flax fiber (which is three times stronger than cotton) is used to make linen in many textile industries due to its natural ability to straighten. Flax fiber is very long also used for the manufacture of decorative fabrics, solid yarn, cordage, and tires [37,44]. Flaxseeds have many nutritional characteristics and contain a high concentration of short-chain omega-3 fatty acids. Flax is mainly cultivated for its fiber but its seeds are edible and can prevent heart disease, cancer, strokes, and diabetes [45,46]. Flaxseed contains 20–25% protein and 40–45% fatty acids. Flaxseed also produces vegetable oils hence also known as ‘linseed oil’, which is an edible oil considered one of the oldest commercial oils and which is also used for pharmaceutical purposes. Flaxseed oil is used as a drying oil agent in many counties in the form of processed solvents. Flaxseed is also used commercially as varnishes and paints. The major producers of flax are Canada, Russia, China, Kazakhstan, the United States of America, and India as shown in Figure 2 [47].

For accumulation and phytoextraction of toxic heavy metals such as lead, arsenic, etc., previous investigations have been conducted [48,49,50,51,52,53,54]. Flax shows a resistance against heavy metals so is an excellent candidate for phytoremediation when grown on metal contaminated soil. Flax is commercially a very important crop used for linseed oil and fiber and significant information is available regarding using flax as a phytoremediation plant in different heavy metal contaminated soils. There has been a growing interest in using this fibrous species for the phytoremediation of different heavy metals. To the best of our knowledge, there has been a lot of investigation using flax for the phytoremediation of heavy metals but no comprehensive review literature is available on this kind of study. In the present literature we will focus on: (a) studies related to phytoremediation of flax, (b) some major characteristics of flax which made it an excellent candidate for phytoremediation, (c) some additional benefits of flax after cultivation on metal contaminated soil, and (d) applications of chelating plant hormones which can increase phytoremediation potential of flax.

## 2. Growth, Morphology, and Habitat

Flax is the oldest fiber crop and grows throughout the world, especially in the colder regions. Flax is cultivated for its fiber, seed, and as ornamental plants in many gardens. Russia is the biggest exporter of flax but the best quality of flax for fiber extraction is grown in Belgium. With the increase in the market demand for fiber and flaxseed, flax cultivation also needs to increase [55,56]. The flax plant is a long, slender, and can grow to a height of 1.2 m (3 ft). Their leaves are simple, glaucous, green, and linear–lanceolate and 3 mm broad and 20–40 mm long [41,57]. Most of the varieties of flax are annual, but some are perennial plants. The flowers have five petals of different colors including white, blue, yellow, and red depending on the species, and are hermaphrodite and hypogynous. The fruit is a round and dry capsule containing many glossy brown seeds composed of five carpels. The roots are fibrous, shallow, and primary roots divide into many lateral roots with numerous branches. Moreover, the roots are straight and can reach a depth of 1.2 m. With diameters ranging from 1.2–2 mm, the flax stem is round to oval. Seeds contain 35–45% oil rich in unsaturated fatty acids (FA) as well as 20–25% protein. The standard FA composition in commercial flax and linseed cultivars is 52–60% α-linolenic (18:3) acids, 2% stearic (18:0), 6% palmitic (16:0), and 13–18% linoleic (18:2) acids, as well as 16–20% oleic (18:1) acid. Flax is cultivated for two major reasons: for obtaining fiber and for its seeds. If flax is cultivated for its seeds, it requires a type of soil similar to the one used for cultivation of wheat, i.e., rich with nutrients. It requires a deep, moist, sandy, well-aged soil containing animal manure. If the flax is cultivated for its fiber, it requires deep, fertile, well-drained soil prepared as when growing vegetables.

Flax thrives best in moist regions with a warm climate—free of late frosts in spring, with sufficient moisture during the growing period and where long, continued rainy spells do not alternate with long, continued dryness. Flax can grow well in cool weather, but sowing depends upon the area. Flax requires a relative humidity of 50–60% with 7 inches of annual rainfall for the best quality yield. Flax can reach its maturity in just 100 days, and it is a good rotational crop [41]. Plants become mature after 100 days or two weeks after the seeds capsules form, and plants turn yellow while the fiber degrades and then plants turn brown. There are two methods for harvesting a flax fiber crop, mechanical and manual methods. The mature seeds are 2 mm long and very rich in oil [55,56]. Some essential plants which are considered good candidates for phytoremediation and their comparison with the flax plants are presented in Table 1.

## 3. Plant Selection Considerations

There is diversity in plants with a vast range in ability to accumulate heavy metal ions. It had been noticed in recent research that metal concentrations of plants grown in the same soil vary from species to species and sometimes between genotypes of plants [66,67]. For heavy metal phytoremediation, plant vegetation should be fast-growing and hardy, easy to cultivate and maintain, and able to transport toxic heavy metals in their above-ground parts as well as through the process of evaporation utilizing an excessive amount of water [27]. Due to the unique properties of pheratophytes, such as deep root systems and fast growth, as well as a large number of stomata in their leaves (high transpiration rate) and the fact that these plants are often native throughout the country of interest, they have been most often selected for phytoremediation in hotter regions of world. For terrestrial species, hybrid poplar is often selected and coontail is selected for aquatic species. In petroleum-contaminated soils, some species like barley, apple, and orange are used because of their ability to secret phenolic and flavonoids compounds [35,68]. Some grasses are also cultivated on tress sites to absorb extra nutrients from the soil. These grasses have tremendously large roots which bind and transform hydrophobic pollutants. These grasses are sown around the trees, which protect them from soil erosion and wind, and remove toxic contaminants from the soil and provide stabilization. Some shrubs also show resistance against heavy metals, such as *Atriplex, Maireana*, and *Glycyrrhiza glabra*, while Sesani and alfalfa grasses are also recommended. Many researchers are also working on metal-contaminated sites where these plants are used for phytoremediation [14,69].

Usually, plants are selected for phytoremediation according to the contaminant of concern as well as the needs of the application. Flax can grow up to 1200 to 1500 mm in length, and its stem is very long, straight, slender, and has some branches on it. The roots system of flax is fibrous, with shallow and lateral root branches, and can grow deep in the soil up to 1.2 m [26,27]. The stem is round and oval with a width up to 2 mm while seeds are rich with nutrients and have many medicinal applications. All these characteristics of flax make it a potential candidate to grow in metal contaminated soils and it can accumulate metal in the order of root > leaves stem > seed > fiber. Thus, flax can be used for phytoremediation and can provide 100% utilizable raw material which have no harmful residue as well as fully biodegradable waste [64,70,71].

Plant species should be adapted according to the conditions of the soil and environment because they are the major source for phytoremediation and the growth of the plant. The use of prestigious plants for novel applications, such as flax, is very favorable due to their different biological activities such as requiring less maintenance, tolerance against stress conditions, fewer environmental effects and human risks, as well as well accumulating plant tissues over heavy metals [42,43,72]. However, particular fibrous plant species may work the best for the remediation of specific contaminants and can be safely used in circumstances where the possibility of invasive behavior has eliminated. Furthermore, beyond phytoextraction, flax can also be used as linseed oil or for fibers at the post-harvest stage. 

## 4. Studies Related to Phytoremediation of the Flax Plant

There were very few studies of flax in heavy metal contaminated soils in the early 1990s. After that, researchers started to cultivate flax in different countries in Europe, especially in Poland. At that time, there were a lot of industries in Europe that polluted the soil by spreading heavy metals. Besides the main products of flax, fiber and flaxseed, researchers also noticed the value of flaxseed in the field of medicine. After that, researchers started working on different heavy metal soils and cultivating flax on these polluted soils. Moreover, many researchers also worked on artificially contaminated soils to find the levels of heavy metals at which flax showed healthy growth and development. The first component of phytoremediation is a biological component, which signifies an existing plant species that shows tolerance and accumulation of heavy metals. These species produce large amounts of above-ground biomass, and metals are transported from below-ground parts to above-ground harvested plant parts. The second component is a technical base component, which contains complex technologies for harvest, growth, regulation of heavy metals by adequate agro-technological actions, and plant protection. Last is the economic component, which comprises the time of extraction, the cost of phytoextraction, and decreases in the cost of the phytoremediation process by adding new products from the heavy metal polluted biomass. There are some studies on flax that was grown under naturally and artificially contaminated soils. We reviewed a few studies associated with diverse heavy metals which are discussed below. Angelova et al. [73] studied flax, hemp, and cotton grown in an industrially polluted region—the Non-Ferrous-Metal Works near Plovdiv, Bulgaria. In their study, they measured the way heavy metals enter the fibrous crops, and the concentrations of heavy metals in plant materials (roots, stems, leaves, seeds, flowers). They also noticed in their study that flax was the crop that most strongly absorbed and accumulated heavy metals from the soil, followed by hemp and cotton. Different heavy metals were accumulated in flax plant organs as follows: roots > stems > leaves > seeds. In their conclusion, they revealed that flax is a suitable crop to cultivate on metal contaminated soils, as it removes considerable quantities of heavy metals from the soil with its root system.

Baraniecki et al. [74] studied the bioremediation potential of flax under Cu, Cd, Pb, and Zn contaminated soils of Bulgaria. They noticed that the high concentration of Cu, Cd, Pb, and Zn were accumulated by flax in the order of root > stem > leaves > seed > fiber. The high concentration of these heavy metals significantly reduced plant growth and biomass. Furthermore, elevated levels of N ratios in the soil did not confirm the expected increase in heavy metal accumulation; the yield of above-ground biomass was higher in polluted soil. Flax removed a high level of these heavy metals from the soil. 

Amna et al. [75] studied (Ni) accumulation of mycorrhizal and non-mycorrhizal flax plants under different concentrations of Ni, i.e., 0 (control), 250, 350, and 500 ppm. Accumulation of metals was higher in mycorrhizal than the non-mycorrhizal plants. Although very high doses of Ni reduced the plant growth and biomass, the mycorrhizal plants showed significant good growth and development when compared with the non-mycorrhizal plants. The summary was that the flax plants could help in the phytoremediation of Ni in artificially contaminated soil, while mycorrhizal plants showed significantly better growth than the non-mycorrhizal plant.

Stritsis et al. [76] studied cadmium (Cd) accumulation in a hydroponic environment of two varieties of flax seedlings (ssp. Usitatissimum and cv. Gold Merchant) under different concentrations of Cd, i.e., 0 (control), 0.1, 0.25, 0.5, and 1.0 μmol L^−1^. Accumulation of Cd was higher at 1.0 μmol L^−1^ and Cd accumulated in the roots as well as the above-ground parts of the plants. The results also depicted that the increase in Cd concentration affects the growth and biomass of the flax seedlings. As the Cd concentration was high in the applied solution, all the growth parameters declined continuously. However, flax seedlings accumulate a high level of Cd in their body parts and show the possibility of phytoremediation.

Saleem et al. [37] conducted a pot experiment using the flax genotype Longya 10 grown Cu-polluted soil which was obtained from a Cu mining area of Hubei province, China, and mixed with natural soil at the ratio of 0:1 (control), 1:0, 1:1, 1:2, and 1:4. The results from this study depicted that the phytotoxicity of Cu reduced the plant growth and biomass while initiating development of reactive oxygen species, which suggested that Cu toxicity caused oxidative damage in the leaves of flax seedlings. They also determined Cu contents from the polluted soil before and after the experiment and noticed that flax could remove a large amount of Cu from the soil and can be used as phytoremediation material for Cu-polluted soils.

Belkadhi et al. [77] studied short term exposure of flax seedlings at different levels of Cd, i.e., 0 (control), 50, and 100 µM. The seeds were soaked in 0 (control), 250, and 1000 µM Cd in Hoagland nutrient solution. Their results depicted that 100 µM caused a significant decrease in the percentages of phosphatidylcholine, phosphatidylglycerol, phosphatidylethanolamine, and changes in other components. However, the application of salicylic acid (SA) played a substantial role in protecting these compounds under Cd-stressed conditions.

Smykalova et al. [49] screened 20 different varieties of flax grown in the Czech Republic. Callus-induction and organogenesis in explants at varying levels of Cd and Zn were provided in the cultivation medium. Their results depicted that increasing levels of Cd and Zn significantly affected the growth of different varieties of flax. In contrast, different varieties accumulated a large amount of heavy metals and thus can be used as a phytoremediation tool for Cd and Zn contaminated soils. Furthermore, they also demonstrated that Llona, Tabor, and Merkur are Zn and Cd tolerant varieties while Venice, Lola, and Jitka are Cd-accumulating varieties and Viltstar is a Cd and Zn sensitive variety.

Najmanova et al. [52] investigated the capacity of flax varieties and cultivars to accumulate Cd under different concentrations: 0 (control), 50, 100, 250, 500, or 1000 M Cd. Their results depicted that most of the Cd was retained in roots, while a little moved to the shoot of the plants. Moreover, a little transformation of Cd concentration through vascular bundles to the above-ground parts showed that flax seedlings could also accumulate Cd in their above-ground parts. 

Hosman et al. [53] conducted a pot experiment using flax plants as a phytoremediator for metal contaminated soil which was artificially spiked with Cd (0 (control), 10, 20, and 40 mg kg^−1^ soil), Pb (0 (control), 150, 500, and 700 mg kg^−1^ soil), and Zn (0 (control), 400, 800, and 1000 mg kg^−1^ soil). Compared to the control, a significant reduction in germination percentage was observed in flax seedlings. Increasing metal concentration in the metal-polluted soil caused a significant increase in the metal uptake by the plant. Flax was able to accumulate a large amount of heavy metals from the soil while the values of bioaccumulation factor (BAF) and translocation factor (TF) were greater than 1 for Zn, and Pb showed that flax is a hyper accumulator species for both Zn and Pb. Contrastingly, BAF and TF were less than 1 for Cd, indicating that flax is a Cd-excluder plant. 

Saleem et al. [12] spiked natural soil with different levels of Cu (0 (control), 200, 400, and 600 mg Cu kg^−1^) and used flax as a tool for phytoremediation under controlled conditions. They proved that flax could tolerate up to 400 mg kg^−1^ Cu while further increments of metal caused a significant reduction in plant growth and biomass. They also determined Cu used the digestion method at different stages of the plant growth, i.e., 35, 70, 105, and 140 days after sowing (DAS). Their results also suggested that in the early stage of the life cycle of the plant, Cu tended to accumulate in the roots and was transported to the shoots at the last stage of the plant life cycle. Furthermore, all the values of BAF and TF were more than 1 which showed that the flax plant is a hyper accumulator species for Cu polluted soils. Hence, based on the following results, the author concluded that the flax plant has considerable potential to revoke a large amount of Cu from the polluted soil.

Belkadhi et al. [54] investigated Cd-treated (100 µM) flax seedlings with different exogenous levels of salicylic acid (SA) (0 (control), 250, and 1000 µM) on H_2_O_2_ initiation, protein composition, and H_2_O_2_-scavenging enzymes. Their results depicted that a high concentration of Cd increased H_2_O_2_ levels and was associated with the augmented activities of guaiacol peroxidase, ascorbate peroxidase, catalase, and superoxide dismutase. The leaves of Cd-free flax seedlings pretreated with SA increased H_2_O_2_ damage to growth and proteins. Moreover, the Cd-treated seedlings primed with SA exhibited a higher Cd concentration in their tissues, which suggests that SA can enhance the phytoremediation of flax in Cd contaminated soil. Uptake and accumulation of different heavy metals by flax plant are presented in Table 2. 

## 5. Value-Added Products

Flax is cultivated in all over the world for many purposes and has many commercial, medical, and other important applications. For instance, the flax plant is cultivated for dual purposes, i.e., flaxseed oil or for fibers. The flax plant is also known for its health benefits. It is beneficial for the human heart as it has essential proteins and fibers, which reduce cardiovascular problems, appetite, and aid in weight control [43,81]. In addition to cardiovascular problems, its seeds are also used for carpal tunnel syndrome and ulcers. Moreover, its seed oil is also used to control blood pressure, rheumatoid arthritis, cholesterol, and many other common diseases in man [82,83]. In addition, flaxseed oil or linseed oil has been ground and pressed to produce natural oil or cooking oil. Flax oil or linseed oil contains alpha-linolenic acid which can be transformed into omega-3 fatty acid, which has many important benefits [47,84]. Similar to flaxseed, flaxseed oil may help lower cholesterol levels. The alpha-linolenic acid in flaxseed oil might play a role in decreasing low-density lipoprotein, or cholesterol [56]. A schematic diagram of flaxseed and flax fiber applications is presented in Figure 3. 

Recently, flax fiber as reinforcement in composites has become popular due to the increasing demand of the market as a result of globalization. Fibers from flax plants are cost-effective, bio-degradable, and exhibit good chemical characteristics [80,85]. Flax was the first fibrous crop that was spun and woven in to the textiles due to its unique properties. Its fibers contain many important pigments such as cellulose, hemicelluloses, wax, lignin, and pectin, which have been studied/reported by many different studies [46,86]. Flax plants have various compositions of pigments as they depend upon plant variety, soil conditions, and growth treatments. Flax fiber has excellent tensile characteristics analogous to glass fibers [46,57]. A flow chart diagram of flax usage is presented in Figure 4.

## 6. Enhancement of Phytoremediation Potential Using Flax Plants

The phytoremediation potential of a plant can be increased by using an exogenous application of metallothioneins or chelating agents. These are essential peptides/proteins involved in metal accumulation and tolerance for any plant species. Plant phytochelatins and metallothioneins have cysteine sulfhydryl groups, which bind with metal ions and form complexes [29,34,87]. Plant species which have a slow growth rate and chemically mediated phytoremediation in a metal stress environment are most effective. However, plant species with huge biomass and fast growth rates are also cultivated in metal stress environments with the help of chelating agents that help to increase the metal availability for such plant species [39,88]. The application of different chelators under metal stress conditions helps to increase metal accumulation and transport within the plant for the development of phytoremediation. In addition to this, the potential of metal absorption in non-hyper accumulator species can be enhanced by the application of such organic chelators. Studies also suggested that different chelating agents are used by the different plant species for extraction and detoxification of heavy metals [26,29,89,90]. Metallothioneins are metal-binding proteins with low molecular weights and encoded by specific genes, which help a plant to tolerate metal stress environments [91,92]. By overexpression of natural chelators (metallothioneins and chelating agents), not only the entrance of metal ions into plant cells but also translocation through the xylem are facilitated.

By using flax, the application of chelating agents such as oxalic acid and ethylenediamine-tetra-acetic acid not only increased biomass and phytoremediation potential but also helped in increase the retting effect on flax by the commercial enzyme products Ultrazym and Flaxzyme (Novo Nordisk) as showed by Nörtemann [93]. Although very few studies are available on the effect of different chelating agents or plant growth regulators of metal contaminated soil [94,95,96,97,98,99,100,101], it is an innovative approach. Still, studies are required in the future to explore more improvements to the phytoremediation efficiency of flax. 

## 7. Conclusions and Future Prospects

Contaminated water and soil with toxic metals is a critical environmental problem, and there is a need to adopt useful methods for the rehabilitation of our environment. Earlier studies showed that flax is appropriate for the phytoremediation of soil and wastewater. Moreover, it can tolerate harsh environments, and uptake, accumulate, and translocate a broad range of diverse metals. Flax is characterized within fiber crops by a genotype for heavy metal tolerance or accumulation. It was also observed that many species are vulnerable to petroleum contaminants but flax has a unique property to tolerate an excessive amount of hydrocarbons and is capable of accumulating a large amount of it in their organs.

In terms of phytoremediation of heavy metals by flax plants, full-scale investigations of the long-term phytoremediation of contaminated sediments are needed to evaluate the influence and the bioavailability of contaminants. These investigations can help researchers to estimate the required time for phytoremediation of a contaminated site. Beside phytoremediation, the flax plant can be used as flaxseed or a fibrous crop. Flaxseed is very important for its flaxseed oil and is thought to treat many diseases in humans, such as cardiovascular diseases and cancer. Flax fiber is naturally very smooth and was very popular in Europe and North America, until cotton overtook flax as the most common plant for making rag-papers. Future research on flax as a phytoremediation tool is very crucial. Long-term research related to the invasiveness and effects of flax on invaded communities may help to develop a feasible control method.

The summarized conclusions of the literature review are as follows:Many studies showed that flax is a hyper accumulator (able to accumulate a larger amount of heavy metals in the aboveground parts then the belowground parts of the plants) species for different heavy metals;A few studies show that flax can also remove petroleum hydrocarbons from contaminated soil. For this property it is very popular in Middle-Eastern countries;Among the different heavy metals, flax can remove Cd the best, and most studies are related to the phytoremediation of Cd, as discussed in detail by Griga and Bjelková [47];Phytoremediation potential and biomass of flax can be improved by using chelating agents or metallothioneins;After phytoremediation, the biomass of flax can be used for the production of value-added by-products such as flaxseed oil or fibers.

## Figures and Tables

**Figure 1 plants-09-00496-f001:**
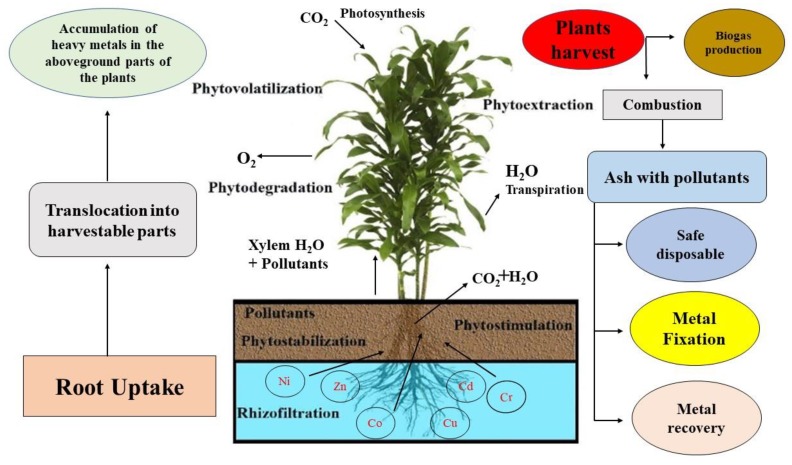
The mechanisms of different types of phytoremediation in plants with different management options of the end product.

**Figure 2 plants-09-00496-f002:**
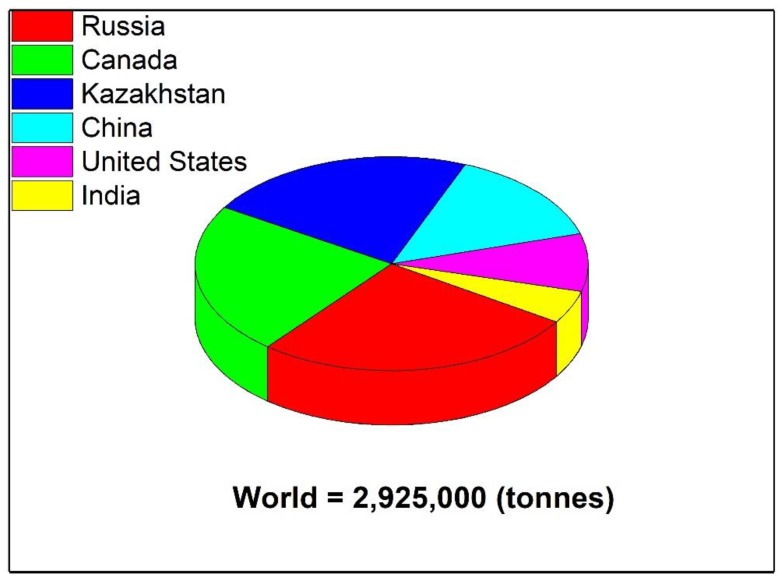
Some major flax producing countries of the world.

**Figure 3 plants-09-00496-f003:**
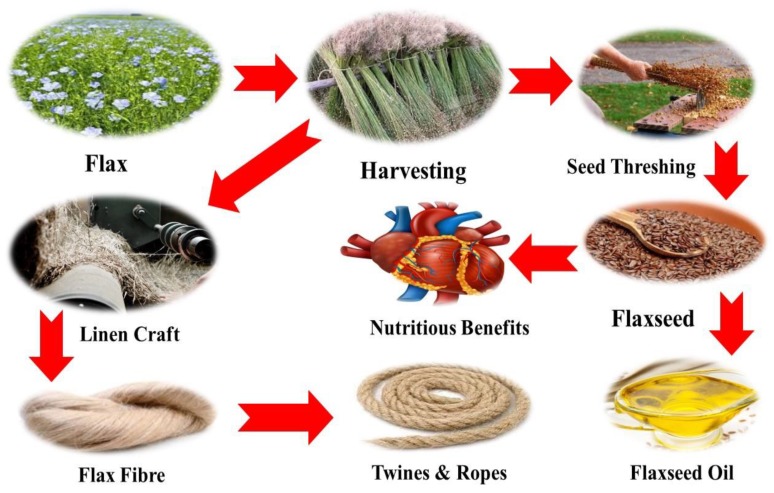
A schematic diagram from the flax field to flax applications.

**Figure 4 plants-09-00496-f004:**
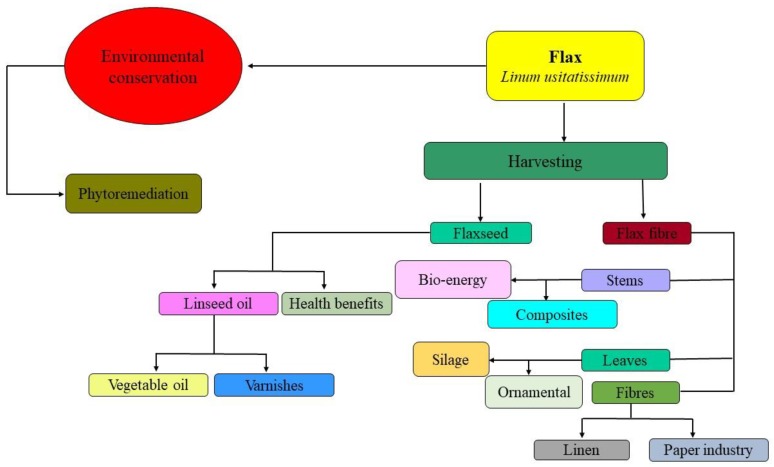
Flow chart diagram of flax usage.

**Table 1 plants-09-00496-t001:** Some essential plants which are considered good candidates for phytoremediation and comparison with the phytoremediation potential and significant characteristics of flax.

Plant Species	Major Characteristics	Phytoremediation Potential	References	Remarks	References
Chinese mustard (*Brassica juncea*)	It is also known as green mustard cabbage, Indian mustard, mustard green, or leaf mustard with edible leaves, stem, and seeds. The vegetative parts of this mustard are consumable, cooked or raw.	It was reported that Chinese mustard has huge biomass and can accumulate an impressive amount of Cd in their above-ground parts.	[58]	Cd is the most frequently studied element in flax plants while the presence of high Cd in soil causes chronic toxicity in humans.	[59]
Chinese ladder brake fern (*Pteris vittata*)	It is native to China and found in many counties of South America and North America. It has a very close resemblance to the swamp fern *Blechnum serrulatum*	The Chinese brake fern has an exceptional ability to accumulate a high amount of As from the soil. It can consist of up to 2% arsenite.	[60]	It was noted that among different heavy metals flax has potential to remove the most amount of As from contaminated soil.	[61]
Pennycress (*Thlaspi Caerulescens*)	Pennycress is a biennial herbaceous plant belonging to the family Brassicaceae. It is a flowering plant and found in Europe and Scandinavia. It also grows on gardens, forest margins, and bare places.	Pennycress can accumulate and tolerate Zn, Cd, and Ni. It frequently occurs on mineralized soils, particularly those with high Zn content.	[62]	When flax grows under different heavy metals (Cd, Zn, and Pb), a progressive decrease in heavy metal content from the soil was observed.	[53]
Barley (*Hordeum vulgare*)	Barley is a member of the grass family, is a cereal crop, and found in the temperate regions of the world. It is used as animal fodder and fermented beer.	Barley can be used as good phytoremediation material in petroleum-contaminated soils and can remove Cd, Pb, Ni, Zn, and Cu from the ground.	[63]	Like very few species (such as sorghum), flax is able to grow on highly hydrocarbon polluted soil, i.e., 40,000 ppm.	[64]
Fescue (*Festuca arundinacea*)	Fescue is a genus of flowering plants and belongs to the family Poaceae. They are evergreen, and herbaceous plants can reach a height of 200 cm. It is used as an ornamental plant and tough grasses.	Fescue has been used for the phytoremediation of Pb and Zn and many other heavy metals and is thus considered as a potential candidate for phytoremediation.	[65]	Flax has huge biomass and can remove 253 mg kg^−1^ Cu from 600 mg kg^−1^ Cu in the soil.	[12]

**Table 2 plants-09-00496-t002:** The capacity of the flax plant to accumulate different heavy metals in the shoots (mg kg^−1^) and roots (mg kg^−1^) when grown in metal contaminated soils.

Metal Type	Metal in Soil	Metal Accumulate by Shoots	Metals Accumulated by Roots	Experiment Type	References
Zn	1008	255	-	Pot	[53]
Cd	41	13	-	Pot	[53]
Pb	704	310	-	Pot	[53]
Cu	617	814	246	Pot	[12]
Pb	1100	332	-	Pot	[78]
Zn	800	116	-	Pot	[78]
Cd	6	49	-	Pot	[78]
Zn	100	26	-	Pot	[79]
Cd	100	190	-	Pot	[79]
Pb	200	104	15	Pot	[80]
Cu	96	31	5	Pot	[80]
Zn	536	213	33	Pot	[80]
Cd	12	9	2	Pot	[80]

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
