# Peer review of "Flax (Linum usitatissimum L.): A Potential Candidate for Phytoremediation? Biological and Economical Points of View"

_plants, 2020, doi:10.3390/plants9040496_

Round 1

Reviewer 1 Report

I was really interested in the review, although I'm not directly involved in this type of research. However, two reference errors among the first two references I consulted (see 1/), shoot accumulation factor are lacking (see 2/) and end-use of contaminated flax must be taking into account.

The paper should be corrected and resubmitted and I will be happy to re-review it.

1/ Figure 2: the unit is lacking. 2925282 what? metric tonnes, km2, acres? The statistic of the production of flax are precise to the unit?

Furthermore, this statistic seems to be taken from two references (Ref 47 and 48, end of page 3). However, ref 47 is wrong. The volume numbers in 2019 for Ecotoxicol. Environ. Saf. are 167-186 (https://www.sciencedirect.com/journal/ecotoxicology-and-environmental-safety/issues).

Ref 48 refers to the effects of phytate on As, P and Fe uptake and growth in As-hyperaccumulator Pteris vittata. No reference to Flax.

2/ Plant's potential for phytoextraction can be estimated by both translocation factor (TF) and shoot accumulation factor (SAF). However, in table 2, these data are not provided. In the text (second paragraph below table 2) we can find a discussion about BAF and TF value but these acronyms are not defined in the text. Such values are clearly needed and the results in table 2 should be listed as the function of the metal, not as the function of the reference.

3/ In part 5, the authors mentioned that after phytoremediation, flax co-products could be marketed. Do the authors seriously think that proteins, fibers or seeds of metal contaminated flax co-products could be commercialized? This paragraph should be removed and replaced by a paragraph dealing with the end-use of contaminated flax.

Author Response

Reviewer # 1

I was really interested in the review, although I'm not directly involved in this type of research. However, two reference errors among the first two references I consulted (see 1/), shoot accumulation factor are lacking (see 2/) and end-use of contaminated flax must be taking into account.

The paper should be corrected and resubmitted and I will be happy to re-review it.

Comment#1: 1/ Figure 2: the unit is lacking. 2925282 what? metric tonnes, km2, acres? The statistic of the production of flax are precise to the unit?

Furthermore, this statistic seems to be taken from two references (Ref 47 and 48, end of page 3). However, ref 47 is wrong. The volume numbers in 2019 for Ecotoxicol. Environ. Saf. are 167-186 (https://www.sciencedirect.com/journal/ecotoxicology-and-environmental-safety/issues).

Ref 48 refers to the effects of phytate on As, P and Fe uptake and growth in As-hyperaccumulator Pteris vittata. No reference to Flax.

Response: Respected reviewer, thanks for my correction and I have corrected according. The references 47 is our previous study which was published in 2019 but came into the issues in 2020 so that the confusion created in the software. However, 48 reference was irrelevant I have changed it with a latest reference in which they explained the import of flax plant in all over the world and I have verified the figure and made a new figure (Fig. 2) from Origin Software which is 3D and easy to understand.

 Although, I used Endnote software and I have updated my software thanks for correction.

Comment#2: 2/ Plant's potential for phytoextraction can be estimated by both translocation factor (TF) and shoot accumulation factor (SAF). However, in table 2, these data are not provided. In the text (second paragraph below table 2) we can find a discussion about BAF and TF value but these acronyms are not defined in the text. Such values are clearly needed and the results in table 2 should be listed as the function of the metal, not as the function of the reference.

Response: Respected reviewer, thanks for your suggestions. BAF and TF values are important in screening hyperaccumulator of a plant’s species. Although we have found a very few literatures in which author calculate BAF and TF values for heavy metals uptake. Even in our previous study we also did not calculate the values of BAF and TF of Cu using flax seedlings.

Although we you are talking about BAF and TF values in the literature which is also our previous study and conducted on Cu contaminated soil.

This is the only literature in which BAF and TF values calculated for the flax for the metal contaminated soil. However, we just gave an idea about metal accumulation in different parts of flax (especially in harvestable parts) that flax has ability to remove a large amount of metals from the soil and can be used as a tool for phytoremediation of heavy metals.

Comment#3: 3/ In part 5, the authors mentioned that after phytoremediation, flax co-products could be marketed. Do the authors seriously think that proteins, fibers or seeds of metal contaminated flax co-products could be commercialized? This paragraph should be removed and replaced by a paragraph dealing with the end-use of contaminated flax.

Response: Respected reviewer, your suggestions are quite right. After the phytoremediation plants are harvested so the idea is that how flax plant can be use in different medicinal and commercial purposes? In this paragraph we have discussed that for what application flax can be used which is the most important application/recommendation. These types of the plants can not be cultivated under very high metal contaminated soil for example Sedum sediforme was able to survive in extreme Cu-toxicity conditions on soils with 5000–16800 mg kg−1 Cu in contaminated soil: Plant Soil 230, 247–256 (2001). The advantages of different plant species under metal contaminated soil (after the phytoremediation) has previously described by many Authors such as Phragmites australis (https://doi.org/10.1007/s11356-019-04300-4).

Thank you for your time and I have improved all my language and English editing in the manuscript and used light green color for the response of your comments.

Reviewer 2 Report

The manuscript of Muhammad Hamzah Saleem et al. reports interesting clarification about Flax phytoremediation,

Long abstract, can be reduced!

English to revise, several spelling mistakes,

The manuscript is clearly written and well organized,

The quality and the content of the graphical "schematic diagram from flax field to flax applications" is rather low

The number of authors is very high?

How do you confirm that the plants used in phytoremediation could be nutritional? 

Author Response

The manuscript of Muhammad Hamzah Saleem et al. reports interesting clarification about Flax phytoremediation,

Comment#1: Long abstract, can be reduced!

Response: Respected reviewer, I reduce some parts of Abstract and just main focus on my idea, novelty and future prospects.

Comment#2: English to revise, several spelling mistakes,

Response: Respected reviewer, thanks for your suggestions. I have revised throughout the manuscript and use also Grammarly and reviewed by other Authors to improve my English standard in better way.

Comment#3: The manuscript is clearly written and well organized, The quality and the content of the graphical "schematic diagram from flax field to flax applications" is rather low

Response: Respected reviewer, I have added a lot of information in the following paragraph and made this figure again as your comment and try to add a lot of information. And write the possible uses of the flax which is related to my study. Thanks for your suggestions.

Comment#4: The number of authors is very high?

Response: Respected reviewer, All Authors participated in writing and reviewing the MS. Although most of Authors are from different countries but we helped each other in writing, reviewing, idea and use of latest software to make graphs and tables.

Comment#5: How do you confirm that the plants used in phytoremediation could be nutritional?

Response: Respected reviewer, as there is not a lot of effect on the growth and morphology of the plant but I has been mentioned in the details that flax after the phytoremediation can be used as dual purposes because the metal concentration is very low as I mentioned to the Reviewer#1 that this concentration of metal is very low and it is described by the other authors that flax can be used to health and nutritious purposes after the phytoremediation. Thanks for your suggestions and I used Blue color for the response of your comments in the MS.

Reviewer 3 Report

The authors are presenting a new approach to an interesting topic. There is no doubt that a wide variety of species can be used for phytoremediation, and the authors do show that flax could be one of them. However, while theoretically sound, the approach does show a variety of problems:

  • the translation into English needs some serious improvements. In the current form the paper is hardly legible. 
  • the idea of the authors that there might be value added products to flax are interesting, but do not coincide with phytoremediation. One has to ask why the authors do only show heavy metal uptake in shoots and roots (but not fibers, seeds etc.). Without any indication as to how much uptake happens in the organs potentially used as "value added" products, the discussion does not make sense - e.g. who would use flax fibers or seeds with high heavy metal content in anything??
  • the authors should delve a bit more into historic literature (e.g. Soviet Union literature) where quite a few of the respective properties are already broadly described.

Overall the paper needs major revisions, and a revision should be resubmitted for review.

Author Response

Comment#1: The authors are presenting a new approach to an interesting topic. There is no doubt that a wide variety of species can be used for phytoremediation, and the authors do show that flax could be one of them. However, while theoretically sound, the approach does show a variety of
problems: the translation into English needs some serious improvements. In the current form the paper is hardly legible.

Response: Respected reviewer, thanks for your suggestions. The English was not good. Although I worked hard to improve the level of my language and also, I used Grammarly software and paper is reviewed by Co-Authors. Hope so this time it will work and revised manuscript is much more attractive than previous one.

Comment#2: the idea of the authors that there might be value added products to flax are interesting, but do not coincide with phytoremediation. One has to ask why the authors do only show heavy metal uptake in shoots and roots (but not fibers, seeds etc.). Without any indication as to how much uptake happens in the organs potentially used as "value added" products, the discussion does not make sense - e.g. who would use flax fibers or seeds with high heavy metal content in anything??

Response: Respected reviewer, after the phytoremediation flax plants is used as many purposes. In my MS I just focus on two main aspects i.e. flaxseed and flax fibre. Although, The metal accumulation does not significantly affected the anatomy of the plants because these types of the species can survive under a little accumulation of metals in the soil. However, I changed the section with
advantages over cultivation of flax and replaced many parts of it which make some sense regarding phytoremediation. In our previous review of literature on jute we also focus on these types of characteristics which can be beneficial to the scientific world. 10.3390/plants9020258.

Comment#3: the authors should delve a bit more into historic literature (e.g. Soviet Union literature) where quite a few of the respective properties are already broadly described.

Response: Respected reviewer, we have mentioned in the MS clearly that Russia is the biggest exporter of flax might be due to weather is suitable for the flax growth. However, I replaced it with  the latest information and made this figure 3D again with Origin Software. Maybe now it looks more attractive than the previous one.

Comment#4: Overall, the paper needs major revisions, and a revision should be resubmitted for review.

Response: Respected reviewer, thanks for your time I made all the figures again and write most of the portion of the MS and highlighted it with different colours and MS is revised by Co-Authors and add latest references and used Grammarly software too. For your response I used red colour in the MS.

Round 2

Reviewer 1 Report

Figure 2. Is the flax production accurate to the nearest ton? Production rounded up to 2’925’000 tonnes could be fine.
Line 285 : what is SA ?
Line 350 : BAF and TF are still not defined in the manuscript.
In their response, no references are dealing with Bioaccumulation factors (BAF). However, even if I’m not a specialist of bioaccumulation, the bioaccumulation factors (BAF) could be calculated by considering metal tissue concentrations with respect to environmental metals concentrations. All the data are in table 2!!

Why did the authors still leave paragraph 5 in the manuscript? The contaminated flax could never be used in any commercial products!
The authors should also discuss about the handling of contaminated flax!

Author Response

Reviewer’s comments

Reviewer # 1

Comment # 1: Figure 2. Is the flax production accurate to the nearest ton? Production rounded up to 2’925’000 tonnes could be fine.

Response: Respected reviewer, I have changed accordingly and make a new figures and write this production tonnes according to your comments.

Comment # 2: Line 285 : what is SA ?

Response: Respected reviewer, SA is the Salicylic Acid. According to the instruction I have written first time full name and then used abbreviation.

Comment # 3: Line 350 : BAF and TF are still not defined in the manuscript.
In their response, no references are dealing with Bioaccumulation factors (BAF). However, even if I’m not a specialist of bioaccumulation, the bioaccumulation factors (BAF) could be calculated by considering metal tissue concentrations with respect to environmental metals concentrations. All the data are in table 2!!

Response: Respected reviewer, you are quite right. BAF and TF should be mention in the MS. I have mentioned this sentence in section 4 which is showing that flax is a hyperaccumulator species for heavy metal contaminated soil. Although, the data in Table 2 is limited and The Authors did not calculate BAF and TF values for the Flax when grown under metal contaminated soil. However, regarding Table 2. We suggest an objective that How much heavy metals can be removed by flax plant or can uptake and accumulate in different body parts.

Comment # 4: Why did the authors still leave paragraph 5 in the manuscript? The contaminated flax could never be used in any commercial products! The authors should also discuss about the handling of contaminated flax!

Response: Respected reviewer, you are quite right flax when grown under metal contaminated soil can never be used as commercial purposes as described there. However, I have changed the dimension of this section and re-write in such as way that what farmers/researchers can obtain from flax plant when grown as fibrous or flaxseed crop. Although, this section is very important regarding my studies that what kind of advantages farmers can got from growing the flax plant on any other soils. Additionally, flow chart and Schematic diagram raise the importance of the flax and also give an attraction to this review article. And some part regarding its handing I have discussed in the Conclusion section. Thanks for your time and I have used red color for the response of your comments.

Reviewer 3 Report

Well revised. Now acceptable for publication.

Author Response

Thank you for your comments and valuable time.